Spatiotemporal distribution of caudal-type homeobox proteins during development of the hindgut and anorectum in human embryos

Tang Xiao Bing 1
Zhang Tao 2
Wang Wei Lin 1
Yuan Zheng Wei 3
Bai Yu Zuo 1 baiyz@sj-hospital.org
1 Department of Pediatric Surgery, Shengjing Hospital , Shenyang, Liaoning , China
2 Department of General Surgery, Affiliated Hospital of Hebei University , Baoding, Hebei , China
3 The Key Laboratory of Health Ministry for Congenital Malformation , Shenyang, Liaoning , China
Esteban María Ángeles
Electronic publication date: 2016 Mar 24
Publication date: 2016
Volume: 4
Electronic Location ID: e1771
Received 2015 Oct 16; Accepted 2016 Feb 16
Copyright: ©2016 Tang et al.
Copyright year: 2016
Copyright holder: Tang et al.
License: This is an open access article distributed under the terms of the Creative Commons Attribution License, which permits unrestricted use, distribution, reproduction and adaptation in any medium and for any purpose provided that it is properly attributed. For attribution, the original author(s), title, publication source (PeerJ) and either DOI or URL of the article must be cited.
License URL: https://creativecommons.org/licenses/by/4.0/

Keywords: Human, Embryo, CDX, Hindgut, Anorectum

Funding: National Natural Science Foundation of China 81470788 This study was supported by the National Natural Science Foundation of China (grant no. 81470788). The funders had no role in study design, data collection and analysis, decision to publish, or preparation of the manuscript.

==============================
Background. The objectives of this study were to determine the spatiotemporal distribution of human caudal-type homeobox proteins CDX1, CDX2 and CDX4 during development of the hindgut and anorectum in the embryo and to explore the possible roles of CDX genes during morphogenesis of the hindgut and anorectum.

Methods. Embryos (89) were cut into sections serially and sagittally. From gestation weeks 4–9, CDX1, CDX2 and CDX4 proteins were detected on the caudal midline by immunohistochemical staining.

Results. During week 4, extensive immunoreactivity of CDX1, CDX2 and CDX4 was detected in the dorsal urorectal septum, urogenital sinus and hindgut. From weeks 5–7, CDX1-, CDX2- and CDX4- positive cells were detected mainly in the mesenchyme of the urorectal septum and hindgut. The levels of CDX2 and CDX4 immunoreactivity were lower compared to CDX1. During weeks 8 and 9, the anorectal epithelium stained positive for CDX1 and CDX4, and the anal epithelium was positive for CDX2.

Conclusions. The CDX proteins are constantly distributed during development of the hindgut and anorectum and exhibit overlapping distribution patterns in the cloaca/hindgut, suggesting they are important in the morphogenesis of the human hindgut and anorectum. CDX genes might be involved in development of the anorectal epithelium after the rectum has separated from the urorectal septum.

Introduction

Anorectal malformations (ARMs) are among the most common human congenital anomalies, occurring in approximately 1/5,000–1/1,500 live births (Van der Putte, 1986), with adverse influences on patient quality of life (Peña et al., 1998; Bai et al., 2000; Levitt & Peña, 2005). ARMs are complex diseases and their etiology, embryology and pathogenesis remain controversial and poorly understood (Wang, Li & Cheng, 2015). ARMs might result from mutations in a variety of genes and the expression patterns of several genes during various stages of gastrulation have helped to clarify the molecular basis of this condition (Van de Ven et al., 2011; Warot et al., 1997; Ramalho-Santos, Melton & McMahon, 2000; Kimmel, Mo & Hui, 2000; Seifert, Harfe & Cohn, 2008; Garcia-Barceló, Chi-Hang Lui & Tam, 2008; Wang, Bai & Wang, 2009; Dravis, Yokoyama & Chumley, 2004).

Caudal-type homeobox (Cdx) genes show highly-restricted expression patterns at the onset of gastrulation, suggesting their involvement in the formation of the digestive tract (McGinnis & Krumlauf, 1992; Silberg et al., 2000; Bonner, Loftus & Wasmuth, 1995). Earlier studies on the spatiotemporal expression patterns of Cdx1, Cdx2 and Cdx4 in rat embryo suggested downregulation of these genes during separation of the cloaca into the rectum and urethra was related to ARM development (Zhang et al., 2009; Tang et al., 2014a; Tang et al., 2014b). Cdx2−∕ − mice displayed severe hindgut abnormalities with failure of colon development and complete terminal blockage (Gao, White & Kaestner, 2009), and Cdx2+∕ −; Cdx4−∕ − mice manifested cloacal septation and anorectal defects, including imperforate anus (Van de Ven et al., 2011). Together these results suggest Cdx genes are related to anorectal morphogenesis in animal models. Distribution patterns of the equivalent human CDX proteins, however, have not been investigated in relation to embryogenesis of the cloaca, hindgut and anorectum, and the involvement of these genes in human cloacal development and their effects on human embryonic hindgut/anorectal development are unknown. This study was designed to determine the distribution patterns of human CDX proteins and their possible roles in hindgut/anorectal morphogenesis. We conducted a systematic study of the spatiotemporal localization of human CDX proteins in normal embryos, with special emphasis on embryonic stages from development weeks 4–9, which represent the crucial time points in human hindgut/anorectal development.

Materials and Methods

Sample preparation

The study protocol was in accordance with the World Medical Association Declaration of Helsinki and was approved by the China Medical University Ethics Committee (no. 200(7) PS14). A total of 89 phenotypically normal human embryos of 4–9 weeks gestation were obtained, with written informed consent, from 22- to 35-year-old women with no history of hereditary disease who were undergoing elective chemically induced/atraumatic curettage termination of unplanned pregnancy (Table 1). Embryos were washed immediately in cold phosphate sodium-buffered saline (PBS pH 7.4) and then fixed in 4% PBS buffered paraformaldehyde (pH 7.4) at 4 °C for 24 h. Samples were dehydrated, embedded in paraffin and then cut sagittally into 4-µm thick sections.

Table 1 Distribution of embryos at different ages.

Gestational age (weeks)	4	5	6	7	8	9	Total	
Number of embryos	9	15	16	18	16	15	89	

Immunohistochemical staining

Endogenous peroxidase activity was blocked by incubation in 3% H2O2 at room temperature for 20 min. Antigens were retrieved by heating the slides in 10 mmol/L sodium citrate buffer (pH 6.0) at 98 °C for 10 min. Sections were treated and incubated with primary rabbit polyclonal anti-CDX1 antibody (LSBio/LS-C180091/48877; 1:200), primary mouse monoclonal anti-CDX2 antibody (LSBio/LS-B4299/38994; 1:50) or primary rabbit polyclonal anti-CDX4 antibody (LSBio/LS-C30413/51929; 1:200) and horseradish peroxidase-conjugated secondary antibody (Santa Cruz Biotechnology Inc., Santa Cruz, CA, USA). Antibodies were incubated in PBS, supplemented with 10% goat serum. Sections were incubated with primary antibodies at 4°C for 16 h and then incubated with secondary antibody for 20 min at room temperature. Immunoreactions were visualized using 3,3P-diaminobenzidine (Sigma, Manchester, UK) as a chromogen. Sections were counterstained with hematoxylin and reviewed independently by two pathologists; the results were agreed by consensus. Negative controls were performed by either omitting the primary or secondary antibody.

Figure 1 Spatiotemporal distribution of caudal-type homeobox proteins during gestation week 4.

During gestation week 4, immunoreactivity specific to CDX1, CDX2 or CDX4 was detected mainly in the dorsal urorectal septum (URS), urogenital sinus (UGS) and hindgut (H). The ventral URS and the cloacal membrane (CM) were negative for CDX1, CDX2 and CDX4. (URS, urorectal septum; UGS, urogenital sinus; CM, cloacal membrane; H, hindgut). Red rectangles in (A, D, G, J) are shown at higher magnification in (B, E, H, K), respectively. Green rectangles in (B, E, H, K) are shown at higher magnification in (C, F, I, L), respectively. Original magnification 100× (A, D, G, J), 200× (B, E, H, K) and 400× (C, F, I, L).

The overall intensity of the immunostaining reaction was evaluated and categorized as “−” to “+”:− negative staining (no colored stain); ±, weak positive staining (light-yellow stain); +, positive staining (yellow-brown stain).

Figure 2 Spatiotemporal distribution of caudal-type homeobox proteins during gestation week 5.

During gestartion week 5, CDX1-, CDX2- and CDX4-positive cells were detected mainly in the mesenchyme of the urorectal septum (URS) and hindgut (H). The cloacal membrane (CM) and urogenital sinus (UGS) were negative for CDX1, CDX2 and CDX4. CDX2 and CDX4 immunoreactivity levels were lower compared to CDX1. (URS, urorectal septum; UGS, urogenital sinus; CM, cloacal membrane; H, hindgut). Red rectangles in (A, D, G, J) are shown at higher magnification in (B, E, H, K), respectively. Green rectangles in (B, E, H, K) are shown at higher magnification in (C, F, I, L) respectively. Original magnification 100× (A, D, G, J), 200× (B, E, H, K) and 400× (C, F, I, L).

Results

In embryos at gestation week 4, a triangular early cloaca was observed at the anterior aspect of the caudal end of the spine. Immunoreactivity specific to CDX1, CDX2 or CDX4 was detected mainly in the dorsal urorectal septum (URS), urogenital sinus (UGS) and hindgut. The ventral URS and the cloacal membrane (CM) were negative for CDX1, CDX2 and CDX4 (Fig. 1).

During week 5, the cloaca was divided into the UGS ventrally and the hindgut dorsally. CDX1-, CDX2- and CDX4-positive cells were detected mainly in the mesenchyme of the URS and hindgut. CM and UGS were negative for CDX1, CDX2 and CDX4. CDX2 and CDX4 immunoreactivity levels were lower compared to CDX1 (Fig. 2).

During gestation weeks 6 and 7, the CM was very thin and elongated. The immunoreactivity levels and distributions of CDX proteins were similar to those seen during week 5 (Figs. 3 and 4).

Figure 3 Spatiotemporal distribution of caudal-type homeobox proteins during gestation week 6.

During gestation weeks 6, the cloacal membrane (CM) was very thin and elongated. CDX1-, CDX2- and CDX4-positive cells were detected mainly in the mesenchyme of the urorectal septum (URS) and hindgut (H). The cloacal membrane (CM) and urogenital sinus (UGS) were negative for CDX1, CDX2 and CDX4. CDX2 and CDX4 immunoreactivity levels were lower compared to CDX1. (URS, urorectal septum; UGS, urogenital sinus; CM, cloacal membrane; H, hindgut). Yellow rectangles in (A, E, I, M) are shown at higher magnification in (B, F, J, N), respectively. Red rectangles in (B, F, J, N) are shown at higher magnification in (C, G, K, O), respectively. Green rectangles in (C, G, K, O) are shown at higher magnification in (D, H, L, P), respectively. Original magnification 40× (A, E, I, M), 100× (B, F, J, N), 200× (C, G, K, O) and 400× (D, H, L, P).

Figure 4 Spatiotemporal distribution of caudal-type homeobox proteins during gestation week 7.

During gestation weeks 7, CDX1-, CDX2- and CDX4-positive cells were detected mainly in the mesenchyme of the urorectal septum (URS) and hindgut (H). The cloacal membrane (CM) and urogenital sinus (UGS) were negative for CDX1, CDX2 and CDX4. CDX2 and CDX4 immunoreactivity levels were lower compared to CDX1. (URS, urorectal septum; UGS, urogenital sinus; CM, cloacal membrane; H hindgut). Yellow rectangles in (A, E, I, M) are shown at higher magnification in (B, F, J, N) respectively. Red rectangles in (B, F, J, N) are shown at higher magnification in (C, G, K, O), respectively. Green rectangles in (C, G, K, O) are shown at higher magnification in (D, H, L, P), respectively. Original magnification 40× (A, E, I, M), 100× (B, F, J, N), 200× (C, G, K, O) and 400× (D, H, L, P).

During gestation weeks 8 and 9, the rectum became separated completely from the UGS and CDX1-, CDX2- and CDX4-positive cells disappeared from the mesenchyme. At the same time, the anorectal epithelium was positive for CDX1 and CDX4, and CDX1 immunoreactivity decreased gradually from the proximal rectum to the anus. The anal epithelium was positive for CDX2 (Figs. 5 and 6).

Figure 5 Spatiotemporal distribution of caudal-type homeobox proteins during gestation week 8.

During gestation weeks 8, the rectum (R) became separated completely from the urogenital sinus (UGS) and CDX1-, CDX2- and CDX4-positive cells disappeared from the mesenchyme. At the same time, the anorectal epithelium was positive for CDX1 and CDX4, and CDX1 immunoreactivity decreased gradually from the proximal rectum (R) to the anus. The anal epithelium was positive for CDX2. (R, rectum). Yellow rectangles in (A, D, G, J) are shown at higher magnification in (B, E, H, K), respectively. Red rectangles in (B, E, H, K) are shown at higher magnification in (C, F, I, L), respectively. Original magnification 40× (A, D, G, J), 100× (B, E, H, K) and 200× (C, F, I, L).

Figure 6 Spatiotemporal distribution of caudal-type homeobox proteins during gestation week 9.

During gestation weeks 9, the anorectal epithelium was positive for CDX1 and CDX4, and CDX1 immunoreactivity decreased gradually from the proximal rectum (R) to the anus. The anal epithelium was positive for CDX2. (R, rectum). Yellow rectangles in (A, D, G, J) are shown at higher magnification in (B, E, H, K), respectively. Red rectangles in (B, E, H, K) are shown at higher magnification in (C, F, I, L), respectively. Original magnification 40× (A, D, G, J), 100× (B, E, H, K) and 200× (C, F, I, L).

Table 2 Spatiotemporal distribution patterns of CDX proteins.

Gestational age (weeks)	CDX1	Protein CDX2	CDX4	
Hindgut				
4	+	+	+	
5–7 (epithelium)	−	−	−	
5–7 (mesenchyme)	+	±	±	
8–9 (epithelium)	Anorectum+	Anus+	Anorectum+	
8–9 (mesenchyme)	−	−	−	
URS				
4	dURS+	dURS ±	dURS ±	
5–7 (epithelium)	−	−	−	
5–7 (mesenchyme)	+	±	±	
8–9 (epithelium)	−	−	−	
8–9 (mesenchyme)	−	−	−	
UGS				
4	+	±	±	
5–7 (epithelium)	−	−	−	
5–7 (mesenchyme)	−	−	−	
8–9 (epithelium)	−	−	−	
8–9 (mesenchyme)	−	−	−	
Notes.

+ Positive staining

± weak positive staining

− negative staining

dURS dorsal URS

The distribution patterns of CDX proteins are given in Table 2.

Discussion

This study showed human CDX1, CDX2 and CDX4 proteins were distributed from gestation weeks 4–9 in a spatiotemporal pattern during embryonic anorectal morphogenesis. During gestation week 4, CDX1, CDX2 and CDX4 were detected in the dorsal URS, UGS and hindgut. From weeks 5 to 7, they were distributed mainly in the mesenchyme of the URS and hindgut. After the anorectum and the UGS opened to the amniotic cavity during week 8, the distribution of CDX1, CDX2 and CDX4 in the mesenchyme decreased, and they were detected in the epithelium of the anorectum/anus. Furthermore, CDX1, CDX2 and CDX4 showed spatially specific distribution patterns in human embryos. They distributed prominently in the dorsal parts of the cloaca that developed into the anorectum, but distributed weakly or almost absent from the ventral part of the cloaca, which develops into the UGS. These results suggest CDX genes might contribute to the development of the cloaca/hindgut and anorectum in the human embryo.

The cloaca is a key feature in the normal morphogenesis of the anorectum (Zhang et al., 2011). Despite their likely complex multifactorial etiology, maldevelopment of the URS and CM is generally thought to be responsible for ARMs (Bai et al., 2004; Qi, Beasley & Frizelle, 2002). The results of this study suggested CDX1, CDX2 and CDX4 were active in the URS during separation of the cloaca from gestation weeks 4–7, but their distribution level decreased after the anorectum and UGS opened to the amniotic cavity in week 8. These findings provide further evidence for the involvement of CDX genes in the maintenance and pattern formation of the URS during development of the hindgut and anorectum. Abnormal expression of CDX genes might impair development of the URS and subsequent morphogenesis of the cloaca/hindgut and could be involved in the development of human ARMs.

Cdx1, Cdx2 and Cdx4 exhibited overlapping expression patterns in the posterior embryo in animal models, and had related functions with regard to their roles in pattern formation of the paraxial mesoderm (Beck, 2004; Lohnes, 2003; Savory et al., 2009; Van den Akker et al., 2002; Van Nes et al., 2006). CDX proteins also exhibit highly overlapping distribution patterns during human cloaca/hindgut development (Fig. 7); CDX1, CDX2 and CDX4 distributed in the same part of the human cloaca from gestation weeks 4–7, indicating CDX genes might have cooperative functions during development of the human hindgut and anorectum. Cross-regulatory interactions might exist among Cdx genes with regard to anorectal development. Van de Ven et al. (2011) showed Cdx1−∕ − and Cdx4−∕ − mice did not develop anorectal defects, whereas Cdx2−∕ − mice did, suggesting Cdx2 has a more prominent morphogenetic role in mice compared to Cdx1 or Cdx4. However, immunoreactivity of CDX1 protein was stronger compared to CDX2 and CDX4 in the cloaca/hindgut and anorectum in this study, suggesting CDX1 might have a more prominent morphogenetic role in the human anorectum compared to CDX2 and CDX4.

Figure 7 A summary schematic of Cdx proteins distribution pattern in rat and human embryo.

The distribution pattern of CDX proteins in humans differ markedly from those of the equivalent proteins in rats. In rats, Cdx1-, Cdx2- and Cdx4-positive cells were located in the cloacal/hindgut epithelium, whereas in human embryos, CDX1, CDX2 and CDX4 proteins located mainly in the peri-cloacal mesenchyme during gestation weeks 4–7 and located in anal/anorectal epithelium during gestation weeks 8–9. (URS, urorectal septum; U, urogenital sinus; CM, cloacal membrane; CL, cloaca; H, hindgut; R, rectum). (Red dots indicate distribution of Cdx1 protein. Yellow dots indicate distribution of Cdx2 protein. Green dots indicate distribution of Cdx4 protein).

CDX genes might be involved in development of the anorectal epithelium. We showed CDX proteins distributed in the anorectal/anal epithelium during gestation weeks 8 and 9. Cdx1 and Cdx2 exhibited transcriptional specificity in the intestine (Grainger, Hryniuk & Lohnes, 2013), and Cdx2 has been shown to be crucial for the expression of signaling molecules, epithelial–mesenchymal interactions and intestinal proliferation patterns (Gao, White & Kaestner, 2009; Grainger, Savory & Lohnes, 2010). Cdx4 is a Cdx2 target gene (Savory et al., 2011). The results of this study and earlier work indicate CDX genes might have a role in development of the anorectal epithelium after the rectum has separated from the UGS.

The distribution patterns of CDX proteins in humans differ markedly from those of the equivalent proteins in animal models (Fig. 7). Cdx1, Cdx2, and Cdx4 are expressed in the developing hindgut endoderm of mice, whereas only Cdx1 and Cdx2 are expressed up to the late gestation and postnatal stages (Beck, 2002). Our earlier studies on the spatiotemporal localization patterns of Cdx1, Cdx2 and Cdx4 proteins in rat embryo suggested Cdx1-, Cdx2- and Cdx4-positive cells were located in the cloacal/hindgut epithelium during development of the hindgut and anorectum (Zhang et al., 2009; Tang et al., 2014a; Tang et al., 2014b) (Fig. 7). In animal models, Cdx proteins are expressed in the epithelium of the cloaca/hindgut, whereas in human embryo, CDX1, CDX2 and CDX4 proteins located mainly in the peri-cloacal mesenchyme (PCM) during anorectal development (gestation weeks 4–7). Asymmetric growth and patterning of the cloacal mesoderm results in division of the cloacal cavity and formation of a genital tubercle (Wang et al., 2011). The cloaca is a key feature in the normal morphogenesis of human anorectum (Zhang et al., 2011). Cloacal membrane (CM) and urorectal septum (URS) play a crucial role on the cloacal embryogenesis (Zhang et al., 2011). These results indicated CDX genes might have a role in dorsoventral patterning of the PCM and suggest misexpression of CDX genes might contribute to maldevelopment of the PCM and subsequent impairment of human hindgut and anorectum morphogenesis.

Conclusions

In conclusion, the results of this study demonstrate CDX proteins distributed throughout the crucial period of hindgut and anorectum development in the human embryo. These proteins exhibit overlapping distribution patterns in the cloaca/hindgut, suggesting they could have a pivotal role in the morphogenesis of the cloaca, hindgut and anorectum, and might be involved in development of the anorectal epithelium. Further studies are required to investigate the role of human CDX genes in anorectal development and their potential involvement in ARMs.

Supplemental Information

Supplemental Information 1 Supplemental Figures of CDX1

Click here for additional data file.

Supplemental Information 2 Supplemental Figures of CDX2

Click here for additional data file.

Supplemental Information 3 Supplemental Figures of CDX4

Click here for additional data file.

Additional Information and Declarations

Competing Interests

Author Contributions

Human Ethics

Data Availability

The authors declare there are no competing interests.

Xiao Bing Tang performed the experiments, analyzed the data, wrote the paper, prepared figures and/or tables.

Tao Zhang analyzed the data, prepared figures and/or tables.

Wei Lin Wang reviewed drafts of the paper.

Zheng Wei Yuan contributed reagents/materials/analysis tools.

Yu Zuo Bai conceived and designed the experiments, contributed reagents/materials analysis tools, wrote the paper, reviewed drafts of the paper.

The following information was supplied relating to ethical approvals (i.e., approving body and any reference numbers):

China Medical University Ethics Committee

No:200(7) PS14.

The following information was supplied regarding data availability:

Data can be found in the Supplemental Information.

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
