# Peer review of "Spatiotemporal distribution of caudal-type homeobox proteins during development of the hindgut and anorectum in human embryos"

_PeerJ, doi:10.7717/peerj.1771_

## Round 0.1 · original submission · Major Revisions

Please amend the manuscript according to the suggestions of the 4 expert reviewers.

Reviewer 1 ·

Basic reporting

The function of caudal-type homeobox genes has been well studied in several animal models, but lack of human data. This manuscript focuses on the expression of caudal-type homeobox genes in human hindgut and anorectum development, which is interesting and will fill the knowledge gap.

According to the author, they have studied 89 human embryos, but the results were not clear. Based on the figures, I can’t see the obvious differences the authors described.

In text, Line 104-111, the authors described figures 1h, 2h, 3h belonged to week8 and 9, but in the figures, it looks h belongs to week7. In addition, figure 1l, 2l and 3l had never been mentioned in the results. Clearly, the manuscript was written in carelessness and I doubt the senior author had carefully read before submit.

The paper can’t be published as it is, and I suggest major revision.

Experimental design

The manuscript use immunohistochemistry to study several CDX proteins localization in different developmental stages of human embryos, the experiment design is simple and well described. Unfortunately, the positive cells are very hard to distinguish from the figures. In figures 1d, 2d, 2f, 2h, 3f, 3h, it’s very hard to see positive cells; I don’t know how the authors get the conclusion from these results. In addition, immunohistochemistry is not a quantitative method to study the expression levels in specific regions. I suggest the authors use qPCR or western blot to study the expression levels of mRNA or protein. I strongly suggest the author show a clear HE stained figure for each stage to show the structure, and reduce the color of counterstain to show clear positive cells. For the differentially expressed region, the authors should use at least 400x photos, and add scale bars in each figure.

Validity of the findings

The positive cells in most of figures are hard to distinguish, so the expression pattern can't be confirmed based on these results.

Reviewer 2 ·

Basic reporting

Review of Tang et al, “Spatiotemporal expression of caudal-type homeobox genes during development of the hindgut and anorectum in human embryo”.

In this manuscript, the authors examine the distribution of three Caudal proteins, CDX1, CDX2, and CDX4, during septation of the cloaca into anorectal and genitourinary sinuses in human embryos. Cdx2 and Cdx4 have been implicated in cloacal septation and hindgut development in mice, making them good candidates to examine in human embryos. There is precious little known about gene and/or protein expression patterns during cloaca development in mice, and even less is known about this process in humans. Thus, this description of CDX patterns in human embryos is important and is likely to be of use to those working in this area of anogenital development.

Experimental design

The design of the experiments is solid. The authors have selected appropriate stages –covering the transformation of a single cloacal sinus into separate anorectal and a genitourinary sinuses – and examined proteins encoded by genes that have been implicated in anorectal malformations (ARMs). Methodological details are sufficient to allow the study to be repeated by readers. The paper is well written and was a pleasure to read.

Validity of the findings

My major complaint about this paper is the quality of the immunohistochemistry; it is difficult to distinguish signal from background in many of the panels. The quality of the sections is excellent, but the clarity of the immunostaining makes it challenging to determine whether the signal is real or is background or even counterstaining. The authors do not mention the use of a counterstain in the methods section, but it appears that a blue counterstain was used. This detail needs to be added to the methods. As for the immunohistochemistry itself, it is difficult to determine whether the lack if clarity is due to high levels of background or suboptimal micrographs.

I have a few recommendations.
1. Firstly, the authors should provide better photos that will allow the reader to see cellular resolution, such as nuclear staining of the CDX proteins. I recommend that they photograph the slides using differential interference contract (DIC, or Nomarski optics). It may be the case that new photos will solve the problem, but I urge the authors to be cautious and to avoid simply taking new photos and resubmitting the paper. This would only be acceptable if the new photos allow one to distinguish signal from background.

2. The biggest problem concerns the mesenchymal staining. In some sections, it looks like every cell is positive, including blood cells. I understand that the authors are working with very rare and precious human embryonic tissue, so I do not want to insist that they embark on large scale repetition of the immunohistochemistry. Rather, I suggest a couple of controls. One control that MUST be carried out is a negative control in which only the primary antibody is omitted but all other steps (all the way through the color detection in activated DAB) should be performed. This will determine the extend of endogenous peroxidase activity of other background staining that is not due to the primary CDX antibodies. A second control that should be considered is a Western blot to show that each antibody recognizes only a single protein in humans. They are using mouse antibodies (including 2 polyclonals) on human tissue, so there this does raise questions about specificity. If each antibody produces only a single band, then they can probably exclude the possibility of cross-reactivity. I would not insist that they do the Western, but I strongly encourage it because it may support the argument that each mouse antibody recognizes only a single protein in humans. Without these data, and with the overlapping domains, the possibility remains that one antibody recognizes other CDX family members in humans.

3. Have the authors considered using a fluorescent secondary antibody to see if the can capture better images with less background?

In summary, I like the paper and am enthusiastic about the authors’ findings, but I also have reservations due to lingering questions about the specificity of the immunohistochemistry and the clarity of the imges.

Additional comments

I think that this will be an excellent contribution to the literature, IF they can resolve the clarify of the staining patterns. We should not hold investigators working on human embryos to the same "experimental" or mechanistic standard that are applied to those working on mice or other modlels. The material is difficult to obtain and descriptive studies like this have tremendous value. They just have to present clearer and more convincing immunostaining.

Reviewer 3 ·

Basic reporting

The manuscript by Tang et al., presents a series of detailed gene expression analyses during human hindgut development. Specifically, the authors reported findings of CDX1 2 and 4, which has been implicated in caudal development. Congenital diseases affecting hindgut development are the major clinical issues but are poorly understood. Genes that are responsible for varieties of clinical presentations including anorectal malformations (ARMs) are largely unknown. Therefore, this is an important area of study that is directly related to human embryogenesis.

Experimental design

immunohistochemical analyses (IHC) of developing human hindgut.

Validity of the findings

1. Because of potential non-specific antibody bindings, it is crucial to include negative controls.

2. In addition to immunohistochemistry, RNA in situ hybridization approaches is needed to validate the IHC findings.

Additional comments

Extensive revision of the manuscript is encouraged.

Reviewer 4 ·

Basic reporting

The submitted article needs improvement in basic reporting in order to effectively communicate the findings such that the data are clear and the conclusions follow logically from the figures presented.

Abstract
line 18 and throughout: use of the word “expression” must be clarified. The authors do not investigate gene expression, and use of the word expression should be unambiguously defined as immunolocalization/protein distribution (for example, line 25 more appropriately explains this concept). Although Cdx genes encode transcription factors and thus protein localization is likely very similar to gene expression, the authors should be clear about the exact experiments that were performed.

Introduction
paragraph 1, sentence 3, lines 42-44: This sentence must be referenced.
sentence 4, lines 44-46: This statement also must be clarified and referenced. The point about gastrulation is confusing; are the authors intimating that improper specification of the endoderm contributes to formation of ARMs?
lines 47-48: same question about gastrulation and ARM
line 60: This study does not determine the expression patterns of the Cdx genes in human anorectal development
line 62: “localization” should be used instead of “expression”

Results
line 99: “URS descended gradually…” the data presented in this study are not able to support claims of morphogenesis. This is a problem throughout the results; the data presented are sufficient to support descriptions of anatomical position at the stages investigated, but this is an example of the authors inferring process from pattern.
other examples: line 104, “elongated progressively”

Figures
The organization of the figures is inconsistent with the results section. One of these should be reorganized in order to make the figures and results parallel; either the figures should be organized by stage instead of by protein, or the results section should discuss the figure panels in the order in which they are presented.

Generally speaking, the magnifications for the figures are too low for the left columns and too high for the right columns to be sufficiently informative. A single column with larger panels or images at an intermediate magnification could better communicate the data.

Why is the overall color balance so varied among stages and samples? For example, in figure 1, panel J appears more green compared to panels D and F.

The red annotations (CM, UGS, H, etc.) are difficult to see, especially, for example, when distinguishing URS and UGS. Either arrows/arrowheads or a different color should be used.

Broad anatomical landmarks, e.g. tailbud/tail and genital tubercle, should be added to the left columns to orient the reader.

The figure legends are redundant. The notable differences among the patterns of Cdx1, 2, and 4 localization are obscured by continued use of identical sentences among each legend. This could be remedied multiple ways; one example is by the use of a brief overall statement of protein localization at the beginning of each figure legend followed by the most important points of the figure that make it different from the other two figures.

Discussion
lines 167-168: “asymmetric growth and patterning of the cloacal mesoderm results in division of the cloacal cavity and formation of a genital tubercle [24]” The literature on cloacal and genital tubercle development do not support the statement that the genital tubercle is formed from cloacal mesoderm morphogenesis. There are necessary links between cloacal and phallus morphogenesis, however, this statement oversimplifies these processes to the point of being incorrect.
line 168: I cannot look into the reference that the authors use for this statement because of inconsistencies in reference style

Experimental design

The submitted article sufficiently meets the requirements of PeerJ in the area of experimental design, with the following exceptions:

Materials and Methods
line 88: why were the sections “reviewed independently by two pathologists” ? Are these individuals authors on the paper? I am confused as to the relevancy of this point.

Why was hematoxylin used as a counterstain, especially in investigating nuclear transcription factors? It seems that a non-nuclear stain, such as Eosin, would have been more appropriate. The hematoxylin dramatically obscures the positive nuclear staining.

Validity of the findings

The general conclusion statements, as in the abstract and introduction, simplify the findings of the paper in a way that suggests more similarities between human and rodent Cdx localization in urorectal development than the authors argue in the last paragraph of the discussion. If there are such marked differences in Cdx localization between mammals and rodents, the data presented here should be sufficiently clear as to support these claims. For example, a summary schematic (properly referencing the rodent data) could succinctly communicate the similarities and differences in Cdx localization between humans and rodents.

With the above exception, validity of the findings in the submitted article sufficiently meets the requirements of PeerJ.

Additional comments

This paper reports important contributions to our understanding of the molecular regulation of urogenital and anorectal development in humans. Upon improving the clarity of data communication in the text and figures, I support its inclusion in the peer-reviewed literature.

minor editorial suggestions:
line 54 punctuation correction Cdx2+/-; Cdx4-/-
line 109 “was” should be used instead of “became”
line 116 punctuation error (comma and period)
line 117: this sentence is redundant with the previous sentence
line 124 change “almost absent, in the” to “almost absent from the”

---

## Round 0.2 · Minor Revisions

Thank you for improving your manuscript which still needs some minor changes as indicated by the reviewer.

Reviewer 4 ·

Basic reporting

Reorganization and adjustments of the figures and figure panels have improved the manuscript. I have concerns about the figures and figure annotations.
1. Please provide titles for the figures.
2. Figure 7, a summary schematic, is of potentially great value. However, the lack of anatomical context and labels makes the information difficult to interpret.
3. The yellow panel labels are almost impossible to see.
4. Red text for anatomical landmarks is not legibile.
5. Scale bar numbers are too small to read.

Experimental design

No comments.

Validity of the findings

In reviews of the first submission, reviewer 1 suggested that the authors show an H&E stained figure for each stage to outline the relevant anatomy. I strongly support this suggestion; anatomical panels as a 4th row below the immunohistochemistry results would allow the authors to omit anatomical labels from the IHC images.

---

## Round 0.3 · accepted · Accept

Thank you for improving your manuscript.